# Polyphosphonate covalent organic frameworks

Ke Xu [1,10], Robert Oestreich[2,10], Takin Haj Hassani Sohi [2,10], Mailis Lounasvuori [3], Jean G. A. Ruthes [4,5], Yunus Zorlu [6], Julia Michalski[2], Philipp Seiffert[2], Till Strothmann[2], Patrik Tholen[7], A. Ozgur Yazaydin [8], Markus Suta [2], Volker Presser [4,5], Tristan Petit [3], Christoph Janiak[2], Jens Beckmann [9], Jörn Schmedt auf der Günne [1] ✉ & Gündoğ Yücesan [2] ✉

Herein, we report polyphosphonate covalent organic frameworks (COFs) constructed via P-O-P linkages. The materials are synthesized via a single-step condensation reaction of the charge-assisted hydrogen-bonded organic framework, which is constructed from phenylphosphonic acid and 5,10,15,20-tetrakis[p-phenylphosphonic acid]porphyrin and is formed by simply heating its hydrogen-bonded precursor without using chemical reagents. Above 210 °C, it becomes an amorphous microporous polymeric structure due to the oligomerization of P-O-P bonds, which could be shown by constant-time solid-state double-quantum $^{31}P$ nuclear magnetic resonance experiments. The polyphosphonate COF exhibits good water and water vapor stability during the gas sorption measurements, and electrochemical stability in 0.5 M $Na_2SO_4$ electrolyte in water. The reported family of COFs fills a significant gap in the literature by providing stable microporous COFs suitable for use in water and electrolytes. Additionally, we provide a sustainable synthesis route for the COF synthesis. The narrow pores of the COF effectively capture $CO_2$.

Covalent organic frameworks (COFs) are microporous materials closely related to metal-organic frameworks (MOFs) and hydrogen-bonded organic frameworks (HOFs)[1–3]. COFs are synthesized via the covalent linkage of organic building blocks to form two or three-dimensional microporous frameworks[2]. Since the first report on boroxine-linked COFs in 2005[2], COF research has been a very active research area due to their tunable porosity and potential for pore functionalization via organic linker design or post-synthesis modifications[4]. To date, many different COF families have been reported in the literature based on their covalent linkage. COFs can be synthesized through a wide range of reactions, including condensation reactions (i.e., boroxine linkage), Schiff base reactions (i.e., imine, hydrazone, enamine, phenazine linkages), click reactions (i.e., triazole linkage), metal-catalyzed coupling reactions (i.e., C-C linkages), etc[5–9]. The vast structural diversity and porosity of COFs enable potential applications such as gas storage, water adsorption, catalysis, water harvesting, $CO_2$ capture, catalysis, photocatalysis, semiconductors, energy storage, and luminescence[5,10–15].

Recently, we have reported on the synthesis of hydrogen-bonded organic frameworks constructed using arylphosphonic acids[16,17]. Due

[1]Department of Chemistry and Biology, Inorganic Materials Chemistry, University of Siegen, Adolf-Reichwein-Straße 2, Siegen, Germany. [2]Institut für Anorganische Chemie und Strukturchemie, Heinrich-Heine-Universität Düsseldorf, Universitätsstraße 1, Düsseldorf, Germany. [3]Young Investigator Group Nanoscale Solid-Liquid Interfaces, Helmholtz-Zentrum Berlin für Materialien und Energie GmbH, Albert-Einstein-Straße 15, Berlin, Germany. [4]INM – Leibniz Institute for New Materials, Campus D22, Saarbrücken, Germany. [5]Department of Materials Science and Engineering, Saarland University, Campus D22, Saarbrücken, Germany. [6]Department of Chemistry, Gebze Technical University, Kocaeli, Türkiye. [7]Technische Universität Berlin, Lebensmittelchemie und Toxikologie, Gustav-Meyer-Allee 25, Berlin, Germany. [8]Department of Chemical Engineering, University College London, London, UK. [9]Institut für Anorganische Chemie und Kristallographie, Universität Bremen, Bremen, Germany. [10]These authors contributed equally: Ke Xu, Robert Oestreich, Takin Haj Hassani Sohi. ✉e-mail: gunnej@chemie.uni-siegen.de; guendog.yuecesan@hhu.de

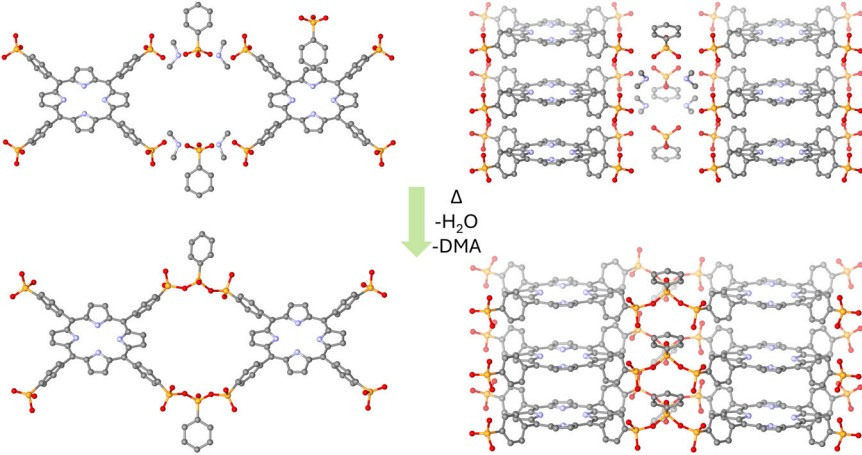

**Fig. 1 | Synthesis route for forming the proposed polyphosphonate-COF GTUB5-COF.** Heating of the precursor HOF GTUB5, which is constructed from phenylphosphonic acid and 5,10,15,20-tetrakis[p-phenylphosphonic acid] porphyrin and HDMA⁺ cations leading to the partially condensed **GTUB5-COF** (proposed structure obtained by DFT calculations, hydrogen atoms are omitted for clarity), and the packing of aromatic porphyrin cores in GTUB5 (above) and GTUB5-COF (bottom).

to their short hydrogen bond distances around ca. 2.5 Å (at shorter than 2.4 Å of donor oxygen-acceptor oxygen pair distances involved in hydrogen bonds are considered to have ca. 50 kJ/mol bond dissociation energy[18]) and formation of multiple hydrogen bonds between the arylphosphonic acid linkers, these HOFs exhibited good stability at 90 °C and 90% relative humidity (RH) and after proton conductivity experiments at 75 °C and 75% RH. Other groups also reported the stability of phosphonic acid HOFs after proton conduction[19]. Our recent unit cell checks on the first batch of GTUB5 (derived from phenylphosphonic acid and 5,10,15,20-tetrakis[p-phenylphosphonic acid]porphyrin) crystals are still providing the same unit cell after 4 years of storage at room temperature and ambient humidity. In the literature, arylphosphonic acids resist thermal decomposition, hydrolysis, and decomposition under ultraviolet light[20]. Expectations on polyphosphonate-COFs, synthesized after condensation of phosphonic acid HOFs, are high stability and structural versatility. They are also expected to generate flexible pores due to flexible P-O-P bonds. Such flexible and stable microporous platforms have the potential to spawn realistic industrial applications to capture gases selectively.

It has been recently shown that branched condensed phosphates exhibit good stability in water[21,22]. Hypothetically, organic diphosphonates and higher polymeric phosphonic acid anhydrides (polyphosphonates) are expected to create even better stability in water as compared to inorganic pyrophosphates as the P-O-P groups are located in between bulky organic moieties (such as the porphyrin linker as shown in this work). Furthermore, hydrolysis of phosphonate esters requires the presence of 37% HCl under reflux conditions to form the corresponding phosphonic acids, indicating a high stability of phosphonic acids in an acidic environment[23]. There is limited information in the literature about the condensation of phosphonic acids. For example, there are a few reports about the ab initio calculations studying the condensation of phosphinic acids and methylphosphonic acid to provide geometry and reaction energies to form dimers, trimers, tetramers, and cyclization[24,25]. There are also only few experimental reports in the literature reporting the condensation of phosphonic acid to form dimers, that is, diphosphonates. For example, a proton-conducting polymer poly(vinyl phosphonic acid) is known to condense forming dimers at higher temperatures, which limits its proton-conducting ability. There is another study by Yücesan et al. that reports the crystal structure of solvothermally condensed phenyl phosphonic acid in acetonitrile at 150 °C, at which the respective product generates a metallomacrocycle with Cu(II) ions and 2,2'-bipyridine[26]. A similar work utilizing condensing agents was reported

later by Zheng using Ag(I) ions[27,28]. The other reports in the literature about the formation of phosphonate dimers, trimers or macrocycles were synthesized using methods, such as ring-opening polymerization of cyclic phosphonates[29,30], In this work, we used our previously reported charge-assisted HOF GTUB5 as a model system to form the polymeric phosphonic anhydride-COF, namely a polyphosphonate-COF formed by condensation of the phosphonic acids phenylphosphonic acid and 5,10,15,20-tetrakis[p-phenylphosphonic acid] porphyrin by simple heating (Fig. 1). Furthermore, we report the structural characterization of a polyphosphonate-COF and explore its thermal, chemical, and electrochemical stability, $CO_2$ capture, water harvesting, and optical properties.

## Results and Discussion

### Synthesis and characterization of phosphonic acid anhydrides

As seen in Fig. 1, the donor-acceptor O-O distances between hydrogen-bonded phosphonic acid moieties of GTUB5 is ca. 2.5 Å, which is slightly higher than 2.4 Å. Hydrogen bonds with shorter donor-acceptor O-O distances than 2.4 Å can have high bond dissociation energies up to 50 kJ/mol[18] (for more detailed information about the hydrogen bond lengths in phosphonic acids, see refs. 16 and 17). Therefore, the phosphonic acid groups in GTUB5 crystals are aligned in a favorable position to initiate the condensation reaction[16,17]. Furthermore, HDMA⁺ (dimethylammonium cations) might act as a condensing agent during the heating. The charge-assisted hydrogen-bonded network of GTUB5 with the presence of fully deprotonated phenylphosphonic acid moieties and HDMA⁺ cations could hypothetically help promote the condensation reaction following the nucleophilic substitution on 5,10,15,20-tetrakis[p-phenylphosphonic acid]porphyrin and fully deprotonated phenylphosphonic acid phosphorus atoms. Thermogravimetric analysis (TGA) coupled with mass spectrometry (MS) of GTUB5 and FT-IR suggests condensation of HOF begins after ca. 130 °C due to evaporation of water and continues until ca. 230 °C. The solid crystals of GTUB5 were heated gradually in an oven to 270 °C to promote the condensation of phosphonic acids to form polyphosphonate units. As detailed below, the GTUB5-COF retained its crystallinity until 210 °C. At this stage, GTUB-COF linkages consist of condensed diphosphonates. The crystals retained their initial color and shape at higher temperatures after heating to 230 °C and 270 °C. Despite the crystalline look of the final product GTUB5-COF, as we discuss below, the samples are X-ray amorphous due to further oligomerization of condensed polyphosphonates.

## Evidence for P-O-P bonds by MAS NMR

Motivated by the amorphous state of the material produced by annealing at 270 °C, we used solid-state $^1$H, $^{13}$C, and $^{31}$P NMR spectroscopy to monitor the structural changes induced by heating between the arylphosphonic acid linkers in GTUB5 under ex-situ conditions, that is, taking samples after annealing steps at different temperatures and characterizing the material subsequently at about 21 °C. Magic angle spinning (MAS) NMR experiments were performed with a 7.04 T magnet spectrometer with Topspin V4.0.5, operating at the frequencies of 121 MHz for $^{31}$P, respectively (See SI for more experimental details). The peaks observed in the $^1$H and $^{13}$C MAS NMR spectra of both untreated HOF GTUB5 and the annealed GTUB5-COF (Supplementary Figs. 1–2, Supplementary Information) were successfully assigned to their corresponding environments (Supplementary Fig. 4,

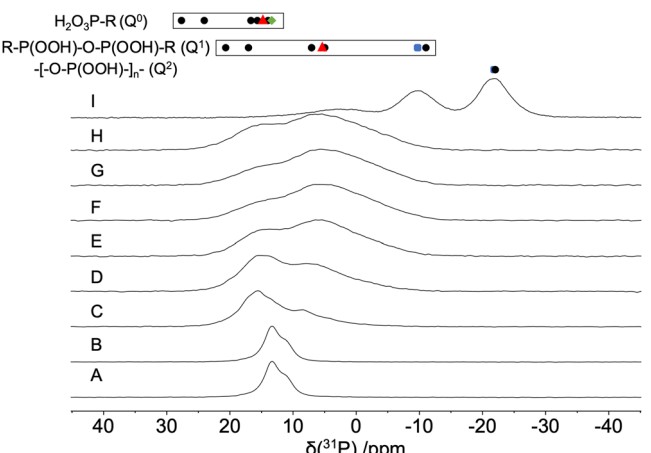

Supplementary Table 1–2, Supplementary Information). These results confirm that no decomposition has happened within the organic moieties of GTUB5 after initial annealing to 220 °C. The $^{31}$P MAS NMR spectra show that the peaks of the untreated crystalline GTUB5 (starting material) broaden gradually at higher temperatures, and a second peak at 5 ppm steadily emerges as the sample is gradually heated to 220 °C (Fig. 2). In the case of inorganic phosphate salts, condensation to pyrophosphates is recognized to induce a shift in resonances, typically shifting them by approximately 10 ppm to lower values than for monophosphates[31,32]. The characterization of protonated phosphates by the isotropic chemical shift is hampered by the substantial influence of the hydrogen-bonded protons[33,34]. To our knowledge, there is no similar correlation tool for phosphonate $^{31}$P shifts. Therefore, we gathered the isotropic chemical shift values for monophosphonates, condensed diphosphonates, and higher condensation products from the literature (Supplementary Table 4, Fig. 2)[31,32]. A similar trend for phosphate groups can also be seen for phosphonate groups. Salts of monophosphonates and hydrogenphosphonate feature a similar isotropic chemical shift range. However, there is a significant overlap between the chemical shift ranges for mono- and condensed-diphosphonate groups. The anisotropic chemical tensor was investigated for differences in the chemical shift anisotropy (Supplementary Table 3, Supplementary Information), but, similar to hydrogen-phosphates, the hydrogen-bonded protons strongly influence the chemical shift tensor, and only insignificant differences were observed.

The magnetic dipole-dipole coupling between neighboring P atoms is a more reliable tool to distinguish condensed diphosphonates from mono-phosphonic acids. The condensation of the R-PO$_3$H$_2$ moieties in GUTB5 to form R-P(O)(OH)-O-P(O)(OH)-R (Fig. 1) moieties upon annealing can be tested by measuring the distances between neighboring P atoms using homonuclear $^{31}$P double-quantum NMR (Fig. 3). In a P-O-P bridge, as in a condensed-diphosphonate group, the shortest P-P distances can be estimated to be of the order of 300 pm[27,28], while in between monophosphonates, the minimum distance would amount to about 460 pm[35,36]. The dipole-dipole coupling relates to the inverse cubic internuclear distance. Therefore, the P-P distances in two-spin systems can be estimated from the magnetic dipole-dipole coupling, for example, by $^{31}$P-$^{31}$P double-quantum NMR experiments with an error of about 10%, including effects by the anisotropic J-coupling[37]. To this end, a homonuclear $^{31}$P-$^{31}$P double-quantum constant-time (DQCT) experiment was conducted on heated GTUB5, in which the peak at 15 ppm under the chosen conditions

**Fig. 2 | A stack plot of $^{31}$P MAS NMR spectra of the sample. a** GTUB5 and GTUB5 annealed at (**b**) 100 °C, **c** 120 °C, **d** 140 °C, **e** 160 °C, **f** 180 °C, **g** 200 °C, **h** 220 °C and (**i**) 270 °C under vacuum. The measurements were performed with a rotor with 2.5 mm outer diameter spinning at 20 kHz (**a**–**h**) or 14.3 kHz (**i**) under a 7.04 T magnet. The box chart indicates the range of the isotropic $^{31}$P NMR chemical shift values of hydrogenmonophosphonates (top, Q$^0$)[31], that of phosphonates with a single bridging O atom including diphosphonates (middle, Q$^1$)[32], and of phosphonates with two bridging O atoms (ring/chain) attached to the observed P-atom (bottom, Q$^2$)[33,34]. The black circles refer to literature values (Supplementary Table 4), the colored symbols refer to the isotropic chemical shift values observed for GTUB5 and GTUB5 annealed at 220 °C and 270 °C.

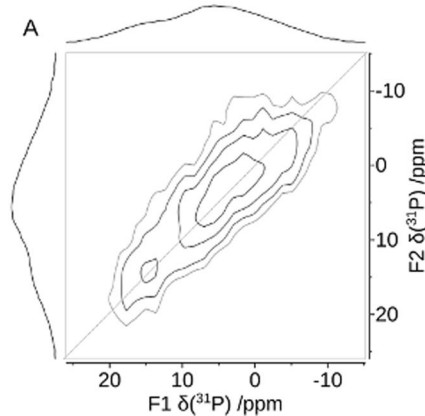

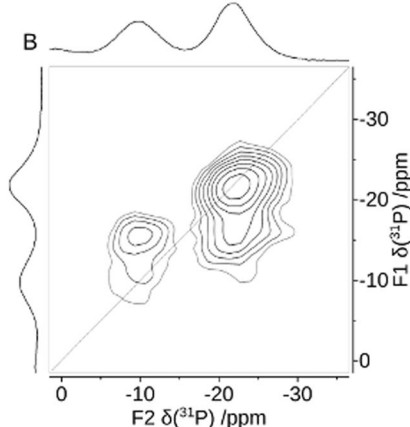

**Fig. 3 | $^{31}$P MAS NMR double-quantum single-quantum correlation spectra MAS NMR spectra of GTUB5.** After annealing at 220 °C (**a**) and after annealing at 270 °C (**b**). The phase-adapted PostC7 pulse sequence[68,69] was used under a magnetic field of 7.04 T at a spinning frequency of $v_r$ = 14286 Hz (left) and 10 kHz (right), respectively. Double-quantum conversion periods for excitation and reconversion period were 1.4 ms (left) and 0.8 ms (right). The drawn diagonal corresponds to the hypothetical position of the signal of an isochronous spin-pair.

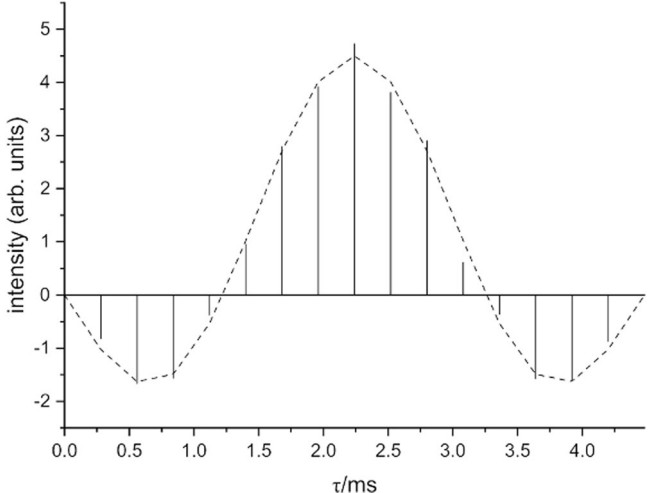

**Fig. 4 | $^{31}$P MAS NMR double-quantum constant-time build-up curve of the pyrophosphonate peak (peak A, Table 2) of the sample GTUB5 after annealing at 220 °C.** The droplines are the experimental data, while the dashed line indicates the fitted data points for a dipole-dipole coupling of -828 Hz. A phase-tuned PostC7 pulse sequence[68,69] was used at a spinning frequency of $v_r$ = 14286 Hz. The constant-time double-quantum conversion period consists of 224 C-elements[38].

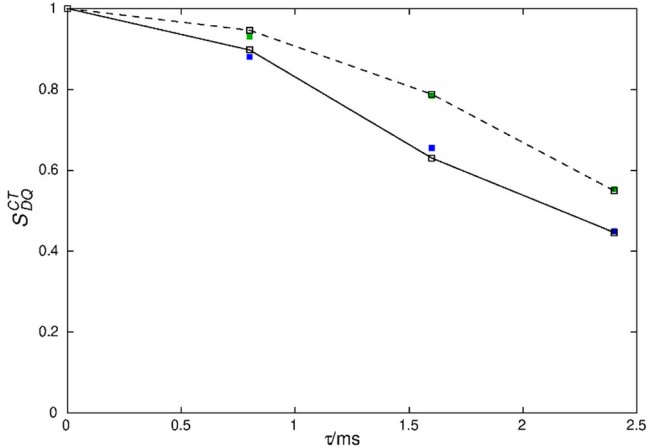

**Fig. 5 | $^{31}$P MAS NMR double-quantum constant-time dephasing curves of the resonance of the sample GTUB5 after annealing at 270 °C.** The experimental double-quantum dephasing ratio $S_{DQ}^{CT}$ for the peaks at −10.1 ppm (filled green squares) and −22.4 ppm (filled blue squares) were fitted by employing numerical simulations of the spin-dynamics with the SIMPSON package[59,60] and yielded effective dipole couplings of $v_{eff}$ of −673 Hz and −1067 Hz, respectively. The constant-time double-quantum dephasing experiment[39] employed a phase-tuned PostC7 pulse sequence[68,69] for excitation, dephasing, and reconversion at a spinning frequency of $v_r$ = 10000 Hz. The double-quantum excitation and reconversion period consisted of 56 C-elements, the double-quantum dephasing period consisted of integer multiples of 14 C-elements, that is, a full supercycle.

shows no zero-crossing while the peak at 5 ppm does (Fig. 4, Supplementary Fig. 5, Supplementary Information)[38]. Based on these observations, the two peaks were unambiguously assigned to condensed-diphosphonate and monophosphonate groups, respectively, which means the annealing to 220 °C only led to a partial condensation.

Double-quantum constant-time (DQCT) conversion curves with well-expressed zero-crossings (Supplementary Fig. 5), as in this case, can be analyzed for the magnetic dipole-dipole coupling by fitting the experimental data in a two-spin approximation (Fig. 4), which for the condensed-diphosphonate group is a good approximation because

further spins are significantly farther away[38]. The analysis revealed a dipole-dipole coupling of $v_{P-P}$ = −828 Hz for the signal at $\delta$ = 5 ppm, corresponding to a P-P distance of $r_{P-P}$ = 2.9 Å within the condensed-diphosphonate group. The bond length includes a small uncertainty of around 10% for the effective coupling constant caused by the anisotropic $^2J(^{31}P,^{31}P)$-coupling, translating into an error of about 0.1 Å[37]. It can be concluded that the presence of a condensed-diphosphonate group can be unambiguously evidenced by solid-state NMR.

From the higher temperature annealing experiments, a GTUB5 sample that was annealed at 270 °C and activated under vacuum for adsorption studies was singled out for further characterization by solid-state NMR, because of the promising mass loss observed in DTA/TG experiments (vide infra). In principle, phosphonate groups could further be condensed forming chain or ring-phosphonates. Regarding the $Q^n$ nomenclature introduced for silicate and phosphate groups, where $n$ corresponds to the number of bridging O atoms attached to a P atom, phosphonates can form $Q^0$, $Q^1$, and $Q^2$ groups but not $Q^3$ groups, which phosphates can do. The tendency to show smaller isotropic chemical shift values the higher the number of bridging O-atoms $n$, is also followed for the GTUB5 compound annealed at 270 °C[22]. Two new resonances are observed at lower shift values (Fig. 2). To prove that the condensation at higher temperatures yields polyphosphonates, again, double-quantum $^{31}$P NMR can be used. Qualitatively, the 2D double-quantum (DQ) single-quantum correlation NMR spectra of GTUB5 annealed at 220 °C and 270 °C show the expected DQ correlation peaks of a condensed-diphosphonate (Fig. 3) and that of a chain phosphonate (Fig. 3) where the peak at −10 ppm is assigned to terminal phosphonate groups ($Q^1$) and the peak at −22 ppm is assigned to chain or ring-phosphonate groups ($Q^2$), respectively. The peak area ratio between the peaks of these two groups of 0.61 indicates an average chain length of 5.3, assuming a chain structure. The assignment via the chemical shift and the occurrence of double-quantum correlation signals can be validated by measuring the effective dipole coupling of the individual $^{31}$P atoms[39]. The effective dipole coupling constant is calculated from the square root of the sum of the squared dipole couplings of an observed nucleus. Because of two bridging P-O-P linkages for $Q^2$ instead of one for $Q^1$, the effective dipole coupling for $Q^2$ can be expected to be 41% larger than that of a $Q^1$ P-atom. The effective dipole couplings were measured with the help of a constant-time double-quantum dephasing experiment (Fig. 5) and indeed amount to −673 Hz for the $Q^1$ peak and −1067 Hz for the $Q^2$ peak[39]. This can be considered an unambiguous confirmation of the above peak assignment and proof of polyphosphonates forming at higher annealing temperatures. The peaks of different $Q^n$ groups observed for GTUB5 annealed at different temperatures were added to the box plot of typical isotropic chemical shift values for further reference (Fig. 2).

## Characterization of polyphosphonate units via FT-IR

The condensation of HOF GTUB5 to GTUB5-COF at higher temperatures was monitored with temperature variable FT-IR spectroscopy. First, the sample was heated to 50 °C, and a spectrum was recorded. Then, the sample was evacuated at 50 °C overnight. This resulted in increased transmittance at 3400 cm$^{-1}$ and 1635 cm$^{-1}$, indicating a loss of water, either structural water incorporated in the HOF or contamination in the KBr used to prepare the pellet (Supplementary Fig. 6).

Then, the sample was maintained in a vacuum as the temperature was increased (Fig. 6a). As the sample was heated, increased transmittance was observed at 2700 cm$^{-1}$, 2450 cm$^{-1}$, 1710 cm$^{-1}$, and 1050 cm$^{-1}$ and assigned to the loss of phosphonic acid groups[40–44]. Decreased transmittance at 970 cm$^{-1}$ was assigned to forming P-O-P linkages[45]. The P-O-H deformation mode in a phosphonic acid is also expected to absorb at around 960 cm$^{-1}$[41,43]. Difference spectra (Fig. 6b)

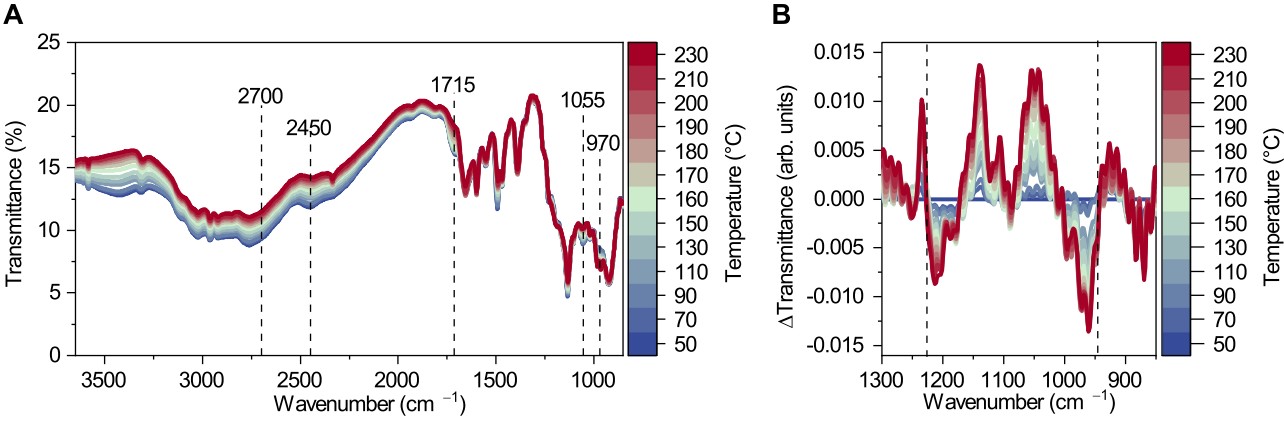

**Fig. 6 | Temperature variable FT-IR. a** FT-IR spectra of HOF GTUB5 during the heating up to 230 °C. **b** FT-IR difference spectra of GTUB5 during the heating.

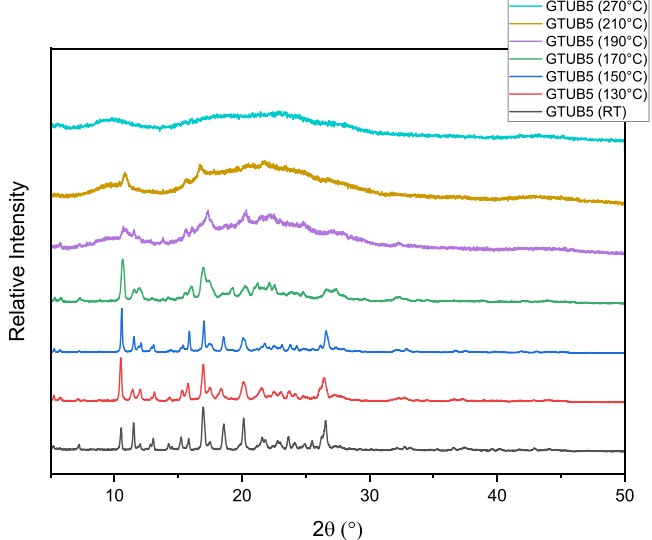

**Fig. 7 | Temperature variable powder X-ray measurements.** Powder X-ray diffractograms were measured from 5° to 50° 2θ for GTUB5-COF at room temperature (20 °C) and heated for 2 h each time at 130 °C, 150 °C, 170 °C, 190 °C, 210 °C, and 270 °C. This figure can be used with Fig. 2. to compare the crystallinity at each temperature-dependent COF phase.

show a bipolar band at this frequency range, supplementary the conclusion that phosphonic acid groups are condensing to form P-O-P linkages. Similarly, difference spectra show a bipolar band around 1230 cm⁻¹ where the P = O bond in both phosphonic acid and condensed-diphosphonate absorbs, and the condensation or the phosphonic acid groups results in a subtle change in the vibrational frequency of the P = O bond (P = O bond is also represented as ⁺P-O⁻, as it is not a classical double bond with π-contribution)[40–45].

## DFT calculations

To confirm the proposed synthesis route, the idealized structure in Fig. 1 was geometry optimized with density functional theory (DFT), as shown in Fig. 1 (bottom). The atom-to-atom distance between hydrogens on the opposing porphyrins in the same pore is about 8.5 Å. To elaborate further on the pore opening of GTUB-COF up on stacking, we placed three geometry-optimized idealized structures on top of each other. We ran a 5 ps ab initio molecular dynamics (AIMD) simulation using the same DFT formalism at room temperature. We observed a gradual constriction of the pore opening, yielding a 5.3 Å atom-to-atom distance at its narrowest section (Fig. 1 and methods section for experimental details).

## Thermal and chemical stability

Although there are several stable COFs in the literature, searching for stable COFs in the presence of water and water vapor and their thermal stability is an ongoing research area[46]. The thermal stability of GTUB5-COF was investigated by initial heating of a sample starting at room temperature up to 270 °C and comparing the powder X-ray diffractograms measured at ambient temperature (Fig. 7). While the crystallinity gradually decreases up to 210 °C, a very similar amorphous diffraction pattern above 210 °C until 270 °C has been observed. We obtained the original unit cell via single crystal X-ray diffraction up to 130 °C, suggesting that the GTUB5 HOF is thermally stable at 130 °C. Although the PXRD data of the original HOF structure pattern is mostly retained up to 190 °C, the single crystal diffraction experiments did not provide a unit cell for the samples heated above 130 °C; due to the gradually decreased crystallinity no further single crystal diffraction measurements could be followed up. As the heated sample at 270 °C retained its crystalline shape and color, we tried 3D electron diffraction to identify the structure of the resulting HOF. Still, getting a data set suitable for structural characterization was impossible. The heated crystals of GTUB5-COF break up in muscle pattern, which is generally seen in glass materials. Figure 7 displays the powder X-ray diffractograms measured at ambient temperature (20 °C), at 130 °C, at 150 °C, at 170 °C, at 190 °C, at 210 °C, and 270 °C from 5° 2θ to 50° 2θ in comparison to each other and the NMR data from Fig. 2. Each time, the same sample was heated for 2 h at the respective temperature.

The heated sample of GTUB5-COF was then added to cold water and boiling water for one hour to test its chemical stability. Due to its amorphous structure, it was impossible to confirm whether GTUB5-COF retained its original structure using PXRD. The crystal shape and color stayed stable in the cold and boiled water after 1 h and did not dissolve or dissipate. The overall structure might have experienced phase transfers due to the presence of flexible P-O-P bonds; we could not confirm this due to the amorphous nature of the GTUB5-COF (Supplementary Fig. 7, Supplementary Information). Furthermore, as larger P-O-P oligomers formed at higher temperatures, the presence of DMA molecules in the pores might have reduced the crystallinity of the resulting COFs. Supplementary Fig. 7 displays the powder X-ray diffractograms of the sample measured after 210 °C heating, after 1 h in cold water, and the activated sample at 270 °C. Although amorphous at this point, a characteristic diffraction pattern can still be identified, thus suggesting the presence of an intact sample.

Furthermore, the constant reproducibility of the water adsorption isotherms for one month also suggests the stability of the reported GTUB5-COF in the presence of water vapor. The water-treated GTUB5-COF at 210 °C and activated GTUB5-COF at 270 °C have two similar broad peaks in their X-ray diffractogram. The disappearance of relatively sharper peaks at the water-treated sample at 210 °C might be due

to the potential release of DMA molecules in the pores, which are miscible with water, increasing the amorphous nature of GTUB5-COF after water treatment. Furthermore, MAS-NMR experiments also proved the presence of linkers at 220 °C.

## X-ray photoelectron spectroscopy (XPS)

XPS measurements were conducted to determine the oxidation state and bonding environment of the nonmetal ions (P, O, N, C) in the heated sample of GTUB5 to 270 °C. The XPS survey spectrum of GTUB5 confirmed the presence of all the elements (C, N, O, and P) of the synthesized material. The survey spectra are shown in Supplementary Fig. 8, and the high-resolution spectra are in Supplementary Fig. 9. The high-resolution carbon 1 s spectrum (C $1s$) can be fitted into 3 peaks with binding energies of 284.5 eV, 284.8 eV, and 286 eV, which indicate the presence of (C=C), (C-C), and (C-N) bonds suggesting the presence of aromatic groups and the porphyrin core. The position of each peak is assigned in Supplementary Table 5. The oxygen spectrum (O $1s$) can be deconvoluted into two peaks positioned at 531 eV and 532.7 eV, which are related to the bridging oxygen (P-O-P) and the non-bridging oxygen (P-O⁻). The positions of these peaks are provided in Supplementary Table 6. The nitrogen $1s$ spectrum (N 1s) consists of three peaks with a binding energy of 400.0 eV, 397.9 eV, and 402.4 eV, which correspond to the presence of (C=N) and (C-N) bonds. The

binding energy peaks of N $1s$ are listed in Supplementary Table 7. The third peak indicates that traces of the solvent DMA can still be detected, which is also confirmed by TGA-MS. The high-resolution spectrum of phosphorus 2p (P $2p$) showed two characteristic peaks at 133.5 eV and 134.9 eV, which were ascribed to (P-C) and (P-O) bonds. The binding energy peaks of P $2p$ are listed in Supplementary Table 8.

## MS-TGA

We coupled thermogravimetric analysis with mass spectrometry to better understand the underlying reaction mechanism. As seen in Fig. 8, the TGA measurement was performed between 30 °C and 600 °C. The green curve displays the number of $H_2O$ ions detected by mass spectrometry, the dark blue line shows the amount of detected dimethylamine, and the light blue line represents the detected DMF. The first step of the TGA curve can be observed at 50 °C, likely due to the loss of the solvent or guest molecules. A second step is noticeable at 230 °C, which is connected to the detection of dimethylamine by mass spectrometry. This affirms the hypothesis that a condensation reaction involving deprotonating the dimethylamine cations and the phosphonic acid groups of GTUB5 occurred. Starting at 390 °C, the third step is connected to detecting dimethylamine and DMF by mass spectrometry. At 390 °C, the complete decomposition of the porphyrin core begins. The formation of DMA and DMF at 390 °C may be explained as pyrolysis products observed after the decomposition of the porphyrin residue.

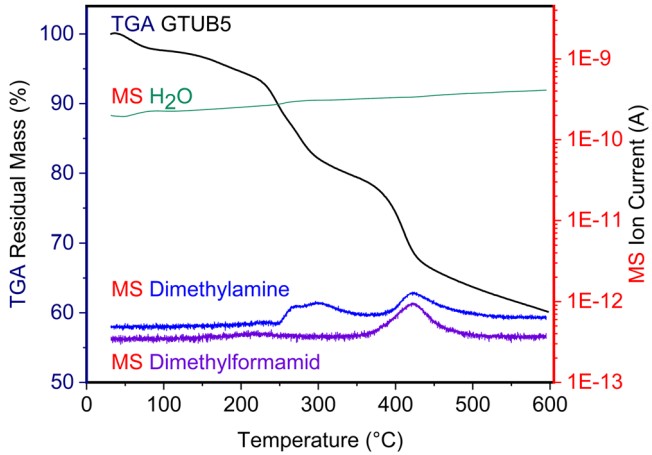

**Fig. 8 | Thermogravimetric analysis.** Mass spectrometry coupled thermogravimetric analysis ($H_2O$, dimethylamine, and dimethylformamide) were measured between 30 °C to 600 °C under a nitrogen atmosphere at a 10 K/min heating rate for GTUB5.

## Electrochemical stability

The reactivity and stability of the GTUB5-COF sample, which was annealed at 270 °C, were investigated in a three-electrode cell setup with a platinum wire and Ag/AgCl (3 M) as counter and reference electrodes, respectively. The open circuit potential (OCP) was measured in an aqueous 0.5 M $Na_2SO_4$ electrolyte for 1 h to evaluate the thermodynamic behavior of the material, followed by linear sweep voltammetry (LSV) in a potential window from −1 V to 2 V vs. Ag/AgCl (3 M) with a scan rate of 1 mV s⁻¹ to assess the electrochemical stability window. To perform the experiments, 50 μL of a 0.5 mg mL⁻¹ aqueous GTUB5-COF dispersion was drop-casted on a glassy carbon electrode (GCE) with a 7 mm² surface area and dried at room temperature. The OCP measurement illustrated in Fig. 9a suggests stability of the GTUB5-COF material in an aqueous neutral environment, exhibiting a noble and stable $E_{OCP}$ = +0.425 V vs. Ag/AgCl, highlighting long-term stability of the material in aqueous electrolytes[47]. The GTUB5-COF was further investigated through LSV, as shown in Fig. 9b. The material exhibited an electrochemical stability window ranging from -0.6 V to +1.3 V vs. Ag/AgCl followed by an oxidation of the aqueous media[48].

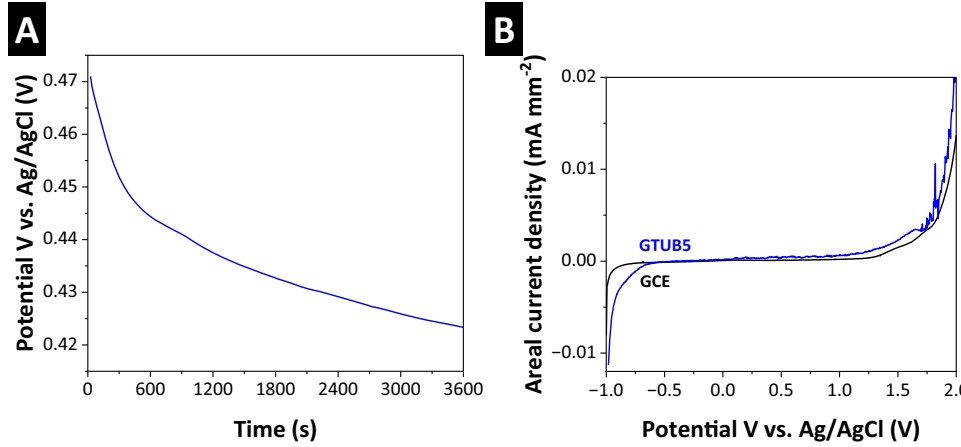

**Fig. 9 | Electrochemical stability of the activated GTUB5-COF sample. a** Open circuit potential (OCP) measurement for 1 h and **b** linear sweep voltammogram recorded at 1 mV s⁻¹ in aqueous 0.5 M $Na_2SO_4$ electrolyte.

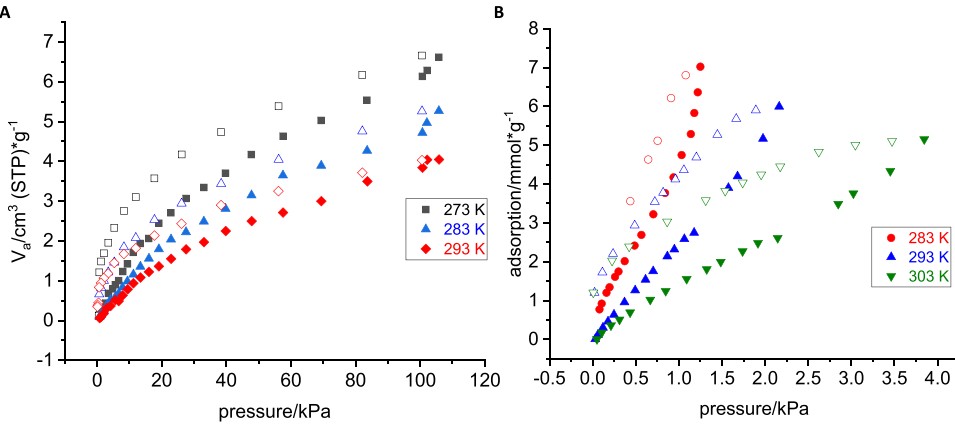

**Fig. 10 | Gas and vapor sorption. a** $CO_2$ sorption at different temperatures (filled symbols adsorption, empty symbols desorption) **b** water sorption isotherms at 283 K, 293 K, and 303 K (filled symbols adsorption, empty symbols desorption).

Such behavior suggests potential application in electrochemical systems such as catalysis, combining a layered compact material, as shown in Supplementary information Supplementary Fig. 10A–F and Supplementary Fig. 11, with favorable electrochemical stability.

## Gas sorption

While COFs have shown potential to capture $CO_2$, there are only a very small number of COFs and MOFs that can capture $CO_2$ in the presence of water vapor, according to a recent comprehensive review article published by Zhao et al.[46,49–51]. To gain more insight into the material and estimate the activation temperature, we performed TGA-MS experiments. According to TGA-MS data, DMA molecules started to leave the pores at ca. 230 °C. Therefore, large needles of GTUB5-COF with sizes between 1 mm and 3 mm (not ground) were activated by heating to 270 °C under vacuum for 2 h to ensure that most of the DMA molecules were evaporated to empty the pores of GTUB5-COF. The activated GTUB5-COF at 270 °C and water-treated GTUB5-COF have an X-ray diffraction pattern aligning with an amorphous material (Supplementary Fig. 7, Supplementary Information). Each $CO_2$ and $N_2$ sorption measurement was repeated three times, and in between each measurement, GTUB5-COF was heated to 60 °C to clear pores from the remaining residual $CO_2$ and $N_2$ gases. Water sorption measurements were performed using a Quantachrome V-STAR4. Between each water sorption measurement, GTUB5-COF was heated to 200 °C for 2 h to remove the residual water from GTUB5-COF's pores. Each of the repeated gas sorption and water sorption measurements provided similar reproducible adsorption isotherms.

Nitrogen sorption measurements at 77 K revealed negligible $N_2$ adsorption capacity and low surface area for GTUB5-COF, which is below 1 m²/g according to BET calculations. In contrast, GTUB5-COF (needles between ca. 1 mm and 3 mm) exhibits notable $CO_2$ and water vapor uptake at room temperature compared to the $N_2$ sorption studies. As we used 1–3 mm long needle-shaped GTUB5-COF sample in the measurements, the error that might originate from fine powder surface area is estimated to be minimal. As seen in Fig. 10a, isotherms at 273 K, 283 K, and 293 K were measured for $CO_2$ adsorption. Much higher $CO_2$ adsorption compared to nitrogen was observed. The distinct adsorption behavior at different temperatures suggests the dynamic nature of GTUB5-COF's porous structure and potential phase transfers due to flexible P-O-P bonds in GTUB5-COF. It is kinetically difficult for $N_2$ to access the pores GTUB5-COF at 77 K. Nevertheless, GTUB5-COF demonstrates that its pores are suitable for $CO_2$ compared to $N_2$. Similar to the $N_2$ adsorption experiments, $CH_4$ also did not show any adsorption, which can be explained by the larger kinetic radius of $N_2$ and $CH_4$ compared to $CO_2$ and water vapor.

The $CO_2$ adsorption isotherms showed a distinct hysteresis over the whole pressure range, indicating favorable interaction between adsorbent and adsorbate. The observed hysteresis in Fig. 10b closely resembles the characteristic features of type H4, as shown by Sing in 2015[52,53]. This observation suggests the presence of small and narrow pores with a polar surface. Following the observed hysteresis, the BET calculation derived from the adsorption isotherm at 273 K gave a $CO_2$-accessible surface area of 47 m²/g and a total pore volume of 0.014 cm³/g. We calculated the enthalpy of adsorption for $CO_2$ by measuring the $CO_2$ adsorption at three different temperatures at 273 K, 283 K, and 293 K via fitting the adsorption isotherms by Langmuir-Freundlich model (Supplementary Fig. 12) and using the Clausius-Clapeyron equation[54]. The enthalpy of adsorption $\Delta H$ of $CO_2$ was calculated to be −27 kJ/mol at a $CO_2$ loading of 0.01 mmol/g. This value is relatively constant over the observed pressure range.

$$\Delta H(n) = -R \cdot ln(p_2/p_1)(T_1 \cdot T_2/(T_2 - T_1)) \tag{1}$$

As seen in Fig. 10 (right), water vapor sorption measurements at 283 K, 293 K, and 303 K showed an adsorption pattern with a slight convex curvature, exhibiting a type II isotherm[52]. This observation also confirms the presence of limited porosity and polar pore surfaces. The enthalpy of water adsorption was also calculated using the same method, giving a value of −21 kJ/mol for a loading of 0.2 mmol/g at a pressure of $10^{-4}$ bar. The load of 0.2 mmol/g was used due to lower accuracy within the low-pressure region for water sorption and according to the available measure points (Supplementary Fig. 12). Unlike $CO_2$, the enthalpy of adsorption for water vapor becomes more negative with increased pressure. The formation of the hydrogen bonds between adsorbed water molecules probably caused the more favorable change in the enthalpy of adsorption for water. Both $CO_2$ gas and water vapor show favorable interaction with the pore surface of GTUB5-COF. Comparability to the enthalpy of carbon dioxide adsorption is complicated due to the distinct characteristics inherent to $CO_2$ gas and water vapor. Nevertheless, the more negative -27 kJ/mol enthalpy of adsorption value for $CO_2$ sorption at lower pressure suggests the potential of GTUB-COF for selective $CO_2$ capture. Furthermore, the repeated consistency of water adsorption isotherms also proves the stability of GTUB5-COF in water vapor. Flue gas contains water vapor, $N_2$, $CO_2$, and $O_2$. To our knowledge, no other works exist that compare the enthalpy of adsorption for $CO_2$ gas and water vapor. Most of the $CO_2$ adsorption with COFs was performed for $CO_2$ against $N_2$. A list of COFs that are stable in water or water vapor with favorable enthalpy of adsorption for $CO_2$ can be seen in Supplementary Table 9.

Supplementary Fig. 13 depicts the diffuse reflectance and luminescence spectra of GTUB5-COF derived from its diffuse reflectance. The localized transitions of the porphyrin entities within the COF characterize it. It reveals the characteristic Soret band at around 350 nm and the Q bands between 430 nm and 700 nm. This suggests the presence of an intact porphyrin core in heated GTUB5-COF, which was activated at 270 °C. The compound shows weak broad-banded fluorescence of the porphyrin units in the near-infrared range, peaking at around 800 nm.

Herein, we report polyphosphonate-COFs. This report presents a straightforward and environmentally friendly method to condense phosphonic acid HOF crystals to synthesize polyphosphonate-COFs in a single step, accomplished by heating the precursor phosphonic acid HOF crystals without additional chemical reagents. Upon gradual heating of the HOF precursor, we initially observed partial condensation of phosphonic acids to generate phosphonic acid anhydride linkages at 220 °C, and after annealing it to 270 °C, we observed the further condensation of phosphonic acids to polyphosphonate chains (or polymeric phosphonic acid anhydride) and rings. We used PXRD, MAS-NMR, FT-IR, XPS, and TGA-MS experiments to monitor the formation of the condensed phosphonate bonds in GTUB5-COF. PXRD data suggests the presence of crystallinity until 210 °C, but crystallinity disappears gradually as the degree of polymerization increases to generate amorphous phases at higher temperatures. The heated sample of GTUB5 retains crystalline COF characteristics until 210 °C. It becomes an amorphous porous polymer with a higher degree of P-O-P bonds at higher temperatures. Based on the MS-TGA data, the HDMA$^+$ cations in GTUB5 donate their protons to the phosphonate groups to promote the condensation of phosphonic acid groups. After the condensation of GTUB5 to form GTUB5-COF, the neutral DMA molecules gradually leave the GTUB5-COF structure at ca. 230 °C, generating void spaces suitable for small molecule capture. GTUB5-COF provides a very dynamic tunable platform leading to a gradually increased degree of P-O-P bond oligomerization at increasing temperatures. The enthalpy of adsorption indicates that GTUB5-COF exhibits higher selectivity for $CO_2$ molecules than water vapor at lower pressures. In contrast, it becomes more selective for water vapor at higher pressures. The reported GTUB5-COF exhibits stability in water and water vapor during the BET measurements, and it is not visually dissolved in cold and boiling water (the dissolved 5,10,15,20-tetrakis[p-phenylphosphonic acid]porphyrin linker produces a very dark red solution).

Furthermore, we have shown that GTUB5-COF is electrochemically stable in 0.5 M $Na_2SO_4$ electrolyte in water, and MAS-NMR data have shown that the linker cores of GTUB5-COF did not chemically decompose at 270 °C. Furthermore, after repeated gas sorption, XPS data have also shown the presence of intact organic linkers and P-O-P bonds. While COF research has yielded diverse families and applications across various fields, a significant and desirable advancement in COF chemistry is the creation of water-stable and $CO_2$-selective COFs. The presented polyphosphonate-COF GTUB5-COF in this work may provide the required stability in water, boiling water, and neutral electrolytes. Both MOFs and COFs are composed of hydrophilic linkages, resulting in a competition between $CO_2$ and water vapor, which mostly favors water adsorption. The small pores exemplify how to utilize the kinetic radii of gases to enhance selectivity for $CO_2$ in the COF research field, and it can address the diverse problems and find industrial use. These findings show GTUB5-COF's potential in controlled gas adsorption and selective $CO_2$ capture, promising directions in sustainable gas storage and separation. Due to the easy synthesis, GTUB5-COF can be easily synthesized on an industrial scale to address a variety of applications. We are currently exploring the full potential of polyphosphonate-COFs, determining their long-term stability in different media, COF thin films, and using longer tethered arylphosphonic acids to improve the surface areas of polyphosphonate-COFs.

## Methods

### Synthesis of GTUB5-COF
Phenyl phosphonic acid and the starting materials for the synthesis of 5,10,15,20-tetrakis[p-phenylphosphonic acid]porphyrin have been purchased commercially from Alfa Aesar and Merck and used without further purification. 5,10,15,20-tetrakis[p-phenylphosphonic acid]porphyrin and HOF GTUB5 were prepared using our previously reported methodology[16]. The condensation of HOF crystals of GTUB5 was performed in oven between 130 and 270 °C.

### Magic Angle Spinning (MAS) NMR
To study the samples GTUB5 heated up to 270 °C, magic angle spinning (MAS) NMR experiments were performed with a 7.04 T magnet spectrometer with Topspin V2.1, operating at the frequencies of 121 MHz for $^{31}$P, respectively. Magic angle sample spinning was done with a commercial 2.5 mm MAS probe head and a self-made 3.5 mm MAS probe head[55]. The chemical shift values refer to the IUPAC reference compound[56]. MAS NMR experiments were also performed with Avance Neo with a 14.1 T magnet spectrometer with Topspin V4.0.5, operating at the frequencies of 242 MHz for $^{31}$P, respectively. Magic angle sample spinning was done with a commercial 1.3 mm MAS probe head. Tetramethyl silane and phosphoric acid were used to reference the shift scale of $^1$H and $^{31}$P NMR, which have the same Ξ values published by IUPAC[56]. The software packages deconv2Dxy-0.4[57] and Simpson-4.2.1[58,59] were used to analyze and fit the NMR spectra. The following equations define the anisotropic chemical shift δ$_{aniso}$ and the asymmetry parameter η:

$$\delta_{aniso} = \delta_{zz} - \delta_{iso} \tag{2}$$

$$\eta = (\delta_{yy} - \delta_{xx})/\delta_{aniso} \tag{3}$$

### XPS
X-ray photoelectron spectroscopy (XPS) was performed with an ULVAC-PHI VersaProbe II microfocus X-ray photoelectron spectrometer. The spectra were recorded using polychromatic aluminum Kα X-ray source (1486.8 eV) and referenced to the carbon 1 s orbital with a binding energy of 284.4 eV. Fitting of the experimental XP spectra was done with the Program CasaXPS, version 2.3.19PR1.0, copyright 1999----2018 Casa Software Ltd.

### DFT calculations
Geometry optimization and AIMD simulation of GTUB-COF were carried out using the CP2K[60] simulation package and employed periodic density functional theory (DFT) calculations within the generalized gradient approximation using the revised Perdew−Burke−Ernzerhof (revPBE-D3) functional and Grimme's D3 dispersion corrections to calculate the energies and the forces[61–63]. The core electrons were treated using the Goedecker-Tetter-Hutter (GTH) pseudo-potentials[64–66], and the valence electrons were expanded using the Gaussian and Plane-Wave (GPW) combined basis sets (double-zeta DZVP along with a plane wave basis set with 400 Ry cut-off energy) as implemented in CP2K/quickstep (version 8.2). The AIMD simulation was run for 5 ps within the Born-Oppenheimer approximation with a time step of 0.5 fs. The temperature was fixed using the Canonical Sampling through Velocity Rescaling (CSVR) thermostat with a time constant of 100 fs[67].

### Transmission electron microscopy
The Tecnai G2 F20 was used to obtain transmission electron micrographs of GTUB5-COF heated to 270 °C.

### FT-IR
FT-IR spectra were recorded with a Bruker 70 v spectrometer (Bruker). For the measurements, 1.6 mg of the HOF powder was mixed with

200 mg KBr and pressed into a pellet with the Atlas Power 15 t hydraulic press (Specac). The pellet was mounted in a vacuum-jacketed variable temperature cell holder (Specac) with pressure <1 mbar. The FT-IR spectra were measured in the transmission mode in the temperature range 50–230 °C.

## Powder X-ray diffraction
Powder X-ray diffraction (PXRD) was performed on a Rigaku Miniflex powder diffractometer in θ/2θ geometry with Cu-Kα radiation (1.54184 Å).

## Optical spectroscopy
Diffuse reflectance and emission spectra were measured on an Edinburgh FLS1000 luminescence spectrometer with 450 W Xe arc lamp, double grating monochromators (Czerny-Turner configuration, blazed at 400 nm in excitation and 500 nm in emission) and a thermoelectrically cooled (−20 °C) PMT980 (Hamamatsu) photomultiplier tube as detection unit. All spectra were acquired at room temperature. Diffuse reflectance was measured using an integrating sphere (diameter 120 mm) setup with the inner surface coated with BenFlect (reflectance >99 % between 350 nm and 2500 nm). All spectra were corrected for the grating efficiency, the lamp intensity, and the wavelength-dispersive sensitivity of the detection unit.

## Gas sorption
Gas sorption measurements were done on the BELSorp-max II by MicrotracBEL Corporation.

## Scanning electron microscopy
A ZEISS GEMINI 500 scanning electron microscope (SEM) was used to obtain the GTUB5 micrographs by applying an accelerating voltage of 1 kV at a working distance of 4.8 mm. Before the measurements, the samples were mounted on an aluminum stub fixed with copper tape.

## Energy dispersive X-ray spectroscopy
A SEM (ZEISS GEMINI 500) coupled with an Xmax detector from Oxford instrument employing an acceleration voltage of 1 kV for imaging and 10 kV for energy dispersive X-ray (EDX) spectroscopy analysis was used for the GTUB5 mapping and the elemental analysis.

## Data availability
Details of synthesis, NMR data, FT-IR data, DFT calculations, XPS data, scanning electron micrographs, transmission electron micrographs, and details of gas sorption studies. All data can be obtained from the corresponding authors upon request. NMR data is accessible at https://fodasi.e-science-service.uni-siegen.de/handle/fodasi/48.

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

## Acknowledgements

K.X., R.O. and T.H.H.S. are equally contributing first authors. We appreciate the support of the Interdisciplinary Centre for Analytics on the Nanoscale of the University of Duisburg. Gündoğ Yücesan and Jens Beckmann thank DFG for funding their research with grant numbers YU 267/2-1, YU 267/9-1, and BE 3716/9-1. M.S. gratefully acknowledges funding by a materials cost allowance of the Fonds der Chemischen Industrie e.V. and financial support by the "Young College" of the North-Rhine Westphalian Academy of Sciences, Humanities and the Arts. K.X., R.O., and T.H.H.S. are equally contributing first authors.

## Author contributions

K.X. performed the NMR characterizations, generated the corresponding figures and contributed to the manuscript. J.S.G. did the analysis of the DQ NMR experiments. R.O. resynthesized the compound, performed gas sorption measurements, organized the TEM measurements at Forschungszentrum Julich, and performed TGA-MS measurements, and wrote the corresponding gas sorption section. T.H.H.S. performed the PXRD work and the chemical stability testing. M.L. performed the temperature variable FT-IR measurements and wrote the corresponding section. J.G.A.R. performed the SEM-EDX analysis and the electrochemical stability testing and wrote the corresponding section. J.M. performed the XPS and wrote the corresponding section. Philipp Seiffert performed the TGA experiments and Till Strothmann designed the TGA-MS experiments. P.T. synthesized reproduced GTUB5 to be used in this work. Y.Z. synthesized the second batch of GTUB5 and contributed to the figures. A.O.Y. performed the DFT calculations and wrote the corresponding section. M.S. performed the optical measurements and wrote the corresponding section. V.P. supervised the work of J.G.A.R. and contributed to the corresponding text. T.P. supervised the FT-IR studies and contributed to the corresponding text. C.J. supervised the gas sorption for BET measurements and edited the manuscript. G.Y. and J.S.G. created the hypothesis, supervised the entire project and wrote the introduction and conclusion and edited the entire manuscript. J.B. provided helpful insights and discussions during the work and contributed to the manuscript. All authors have approved the final version of the manuscript.

## Funding

## Competing interests

Gündoğ Yücesan has filed a patent for the condensed microporous phosphonic acids. The other authors declare no competing interests.
