## [Peer Review File · Nature Communications]

REVIEWER COMMENTS

Reviewer #1 (Remarks to the Author):

The Paper by Yücesan and Schmedt auf der Günne et al. reports the condensation of a HOF to a COF using phosphonates. The phosphonate HOF was already published by them and is now readily converted into a corresponding new phosphonate COF by heating. The authors develop and apply several methods to assess the nature of the new COF and provide diverse evidence for the stepwise condensations taking place. The appealing feature of their approach is that it should now be quite straight-forward to introduce modifications to their frameworks by simply exchanging phosphonates as additives followed by heating. The stability of the new COFs is high as demonstrated by them in several applications, including electrochemistry under quite harsh conditions.

Overall, I think this is a very high quality study that deserves publication, after clarification of a few issues.

- 1) pyrophosphate is (usually) reserved for inorganic pyrophosphate with no modifications, so the authors may want to consider to name these structures as diphosphonates (or phosphonic acid anhydrides)
- 2) The authors refer to branched inorganic pyrophosphates and again I would not use this term as a 2-phosphate unit cannot branch. It would be better to refer to these structures as branched phosphate anhydrides or ultraphosphates
- 3) The statement "Hypothetically, organic pyrophosphonates are expected to create better stability compared to inorganic pyrophosphates". I do not understand this argument without explanation (and it does not have a reference), also again because I am not sure what is meant with inorganic pyrophosphates (is there a plural?). Both are anhydrides and so both can be principally hydrolysed. Furthermore, I would not say "create" in this context.
- 4) The statement: "there is limited information on the condensation of phosphonic acids" is not correct. I think there is a lot of information and the authors should make clearer that they (probably) mean: without using chemical reagents, just heat. That is not fully clear here and then the statement is not correct. Also, there could be a little discussion as to why that is important for the COF formation, or would the addition of condensing agents be an option (if so, it would be nice to include a test reaction of HOF to COF transformation using EDC for example)
- 5) Related phosphonic anhydride structures have been discussed in Chem Eur J, 2023, 29, e202302400; a list of potentially useful ³¹P NMR chemical shifts was published in a table in Acc. Chem. Res. 2021, 54, 4036
- 6) The authors could test if primary amines are capable of reopening the framework. Often, linear oligophosphates with modifications in the periphery can be nucleophilically opened by primary

amines. It is interesting to see that DMA does not do that apparently, but a primary amine might be used to cleave the phosphonic anhydrides again (then, acid treatment would cleave the P-N bonds, and the HOF might be reformed)

7) Scheme 1 shows the potential structures. I find it hard to imagine (and it is only mentioned in the text), how cyclic structures may form in the framework. If the authors would show a proposed structure, that would be much easier to follow (I would also not think that a cyclic modified phosphonate would be stable in the presence of DMA).

8) Figure 2 (and elsewhere): I would not write P(OOH), because that would indicate the presence of a hydroperoxide; rather P(O)OH

9) Figure 8 has a bad resolution

10) Table 2 again contains P(OOH)

Overall, if the issues above can be addressed/corrected I would recommend of publishing this nice work in Nat. Comms.

Reviewer #2 (Remarks to the Author):

The article of Xu et al. introduces a new type of COF, specifically polyphosphonate-COFs, constructed via P-O-P linkages. The synthesis method is notable for its simplicity, involving a single-step condensation reaction of a pre-existing hydrogen-bonded organic framework (HOF) without the need for additional chemical reagents. This method represents a straightforward and environmentally friendly approach to COF synthesis. The resultant GTUB5-COF demonstrated impressive water, vapor stability, and exceptional electrochemical stability, marks a significant advancement in COF materials, particularly for applications in gas storage and separation. The data mostly affirm the authors' main conclusions, but there are some instances where the experimental results are not adequately interpreted. Overall, an important study worthy of publication in Nature Communications before addressing the following issues.

1. The amorphous nature of the synthesized GTUB5-COF posed challenges for X-ray diffraction analysis, which limited the detailed structural understanding of the material. Additional characterization techniques such as X-ray photoelectron spectroscopy (XPS) and advanced electron microscopy are needed. These could provide deeper insights into the chemical states and the morphological details of GTUB5-COF, complementing the findings from NMR and FTIR analyses.

2. Given the amorphous nature of GTUB5-COF, computational modeling might offer insights into its probable structure and properties. Molecular dynamics (MD) simulations or density functional

theory (DFT) calculations could help understand the gas adsorption capacities and selectivity, as well as the material's stability under various conditions.

3. Perform long-term stability studies of GTUB5-COF under practical operational conditions, including variations in temperature, pressure, and exposure to different gases and liquids.

4. Conduct a comparative study with other COFs or porous materials targeting similar applications. Such comparisons could highlight the unique advantages or potential drawbacks of GTUB5-COF, providing a clearer context for its practical utility.

5. It is better to elaborate on specific application scenarios or case studies where GTUB5-COF could be particularly advantageous. For instance, detailing its use in a specific industrial gas separation process or in environmental remediation could underscore its practical value and spur interest in further research and development.

6. The synthesis method appears scalable and environmentally friendly, but a detailed cost analysis and scalability study would be beneficial for assessing the potential for industrial applications. Scalability is a critical factor for transitioning from laboratory research to industrial applications, and addressing this could significantly enhance the paper's impact.

7. Authors should carefully consider their wording, especially in the introduction. For example "The thermal and chemical stability and permanent porosity observed in some COF families opened many potential applications such as gas storage, water adsorption, catalysis, water harvesting, CO₂ capture, catalysis, photocatalysis, semiconductors, energy storage, luminescence."

Reviewer #3 (Remarks to the Author):

Xu et al. proposes a new chemistry for COF making, with an indirect route of HOF to COF conversion. It is not every day we see a new class of covalent linking is presented. The challenge, however, is to verify the structures. I think the findings have value but need significant revisions. Here're the most important ones:

1. The main issue with the manuscript is the claim for COF. Although there is no prerequisite of being crystalline to be considered a COF, the common understanding of the field is to expect some degree of crystallinity. Even if a paper reporting new COFs is published without any evidence of order, the community will disregard the findings and also discredit the authors. I think it's important to either reframe the structures as something else or try further to get crystallinity.

2. If there is no crystallinity observed, it might not be a bad idea to widen the structure bandwidth. For example, why can't the porphyrin-based monomers condense with each other to make framework structures?

3. The HOF to COF transition experiment shows retention of the grain morphology but not the order. This might be due to the oligomers forming, as the condensation with counter cations must be sterically challenging. The disruption is enough to post disorder but not enough to change the form of the particle.

4. The CO₂ uptakes are not exceptional. It's probably better to focus on water absorption or H₂S if possible. One evidence is that the H₂O uptake is an order of magnitude higher than CO₂. There are hints on proton conductivity with these materials, but the authors didn't push forward. It might not be a bad idea to focus there.

Formatting:

1. It is hard to navigate with the figures being at the end. If it is revised for another round of reviews, I would prefer seeing them where they first appear.

2. "using a simple heating in the literature." probably meant "by simple heating."

3. It is better to put both chemical structures and crystal structures in the Scheme 1. Also, I believe Schemes are not allowed in Nature.

4. What do you mean by "ca. 2.5 Å (hydrogen bonds with shorter donor-acceptor O-O distances than 2.4 Å)"? 2.5 Å is longer than 2.4 Å?

5. There are overwhelming details of equipment in the main text. They should be moved to supplementary information.

Response to Reviewer Comments

We are delighted with the valuable comments from the three reviewers and appreciate their suggestions for accepting the article in Nature Communications. In response to the reviewers' comments, we have added two new authors to the article who performed XPS, TEM and DFT calculations and included transmission electron micrographs. Below, you will find our responses to the reviewers' comments **in red**.

Our Responses to Reviewer #1

Comment 1: pyrophosphate is (usually) reserved for inorganic pyrophosphate with no modifications, so the authors may want to consider to name these structures as diphosphonates (or phosphonic acid anhydrides).

Our response: As Reviewer 1 suggested, the term pyrophosphonates has been replaced by condensed-diphosphonate. The term phosphonic acid anhydride is not specific enough as it includes all kinds of polyphosphonates.

Comment 2: The authors refer to branched inorganic pyrophosphates and again I would not use this term as a 2-phosphate unit cannot branch. It would be better to refer to these structures as branched phosphate anhydrides or ultraphosphates:

Our response: We have corrected the sentence to "... branched condensed phosphates" as in the original article.

Comment 3: The statement "Hypothetically, organic pyrophosphonates are expected to create better stability compared to inorganic pyrophosphates". I do not understand this argument without explanation (and it does not have a reference), also again because I am not sure what is meant with inorganic pyrophosphates (is there a plural?). Both are anhydrides and so both can be principally hydrolysed. Furthermore, I would not say "create" in this context.

Our response: The respective diphosphonate groups are partially protected by bulky porphyrin rests, which suggests that it is difficult for water, acids, and bases to assess the P-O-P units. To provide clarity, we explained it with additional text in the manuscript.

Comment 4: The statement: "there is limited information on the condensation of phosphonic acids" is not correct. I think there is a lot of information and the authors should make clearer that they (probably) mean: without using chemical reagents, just heat. That is not fully clear here and then the statement is not correct. Also, there could be a little discussion as to why that is important for the COF formation, or would the addition of condensing agents be an option (if so, it would be nice to include a test reaction of HOF to COF transformation using EDC for example) We can look for some more references regarding the formation of COFs.

Our response: We corrected the text as mentioned below. The amount of work we cited on the chemical transformation of phosphonic acids into anhydrides and related theoretical studies is already substantial enough for a mini-review, but they consist of random reports. Upon re-examining the literature, we could not find experimental work specifically related to the condensation of phosphonic acids without using chemical reagents. Zheng's work, which we have cited, already utilizes condensing agents to synthesize phosphonic acid anhydrides. Furthermore, HDMA⁺ cations also function as a condensation agent. We updated the manuscript accordingly.

The key advantage of our work is that it achieves condensation through a simple heating process, without the need for an additional condensation agents. This is the first example of condensing phosphonic acids by heating and polymerizing them at higher temperatures. We also discussed the potential role of HDMA⁺ cations in the pores, promoting the condensation reaction in our work. We have clarified this distinction in the manuscript and expanded the discussion on why this approach is important for COF formation.

Comment 5: Related phosphonic anhydride structures have been discussed in Chem Eur J, 2023, 29, e202302400; a list of potentially useful ³¹P NMR chemical shifts was published in a table in Acc. Chem. Res. 2021, 54, 4036

Our response: We have included these references and updated the chemical shift tables accordingly.

Comment 6: The authors could test if primary amines are capable of reopening the framework. Often, linear oligophosphates with modifications in the periphery can be nucleophilically opened by primary amines. It is interesting to see that DMA does not do that apparently, but a primary amine might be used to cleave the phosphonic anhydrides again (then, acid treatment would cleave the P-N bonds, and the HOF might be reformed)

Authors' Response: Thank you very much for this valuable suggestion. Currently, we are conducting long-term chemical stability tests on these materials with another group of PhD students, we would also like to test this very interesting hypothesis in our new work. As Robert Oestreich and Xu Ke will be defending their PhD theses soon, we plan to incorporate this suggestion into a new, more detailed, and focused article. This future work will include proton conductivity and photoluminescence studies, post-synthetic modifications, and further stability tests. At this stage, our current study focuses on the stability in neutral electrolytes, pH, and water. The stability of the reported COFs in water is an important feature of the presented COFs, as many COF families readily dissociate in the presence of water and water vapor.

Comment 7: Scheme 1 shows the potential structures. I find it hard to imagine (and it is only mentioned in the text), how cyclic structures may form in the framework. If the authors would show a proposed structure, that would be much easier to follow (I would also not think that a cyclic modified phosphonate would be stable in the presence of DMA).

Our response: Thank you very much for this valuable comment. We have now prepared a new Scheme 1 that better explains the structures. We already have the crystal structure of the precursor HOF. DFT calculations have also shown that the proposed condensed structure is stable. Furthermore, we have collected XPS data that have proven the presence of P-O-P bonds at 270 °C. Nucleophilic substitution on phosphonate phosphorus is challenging since the high dissociation barrier of the P-C bonds in phosphonates has to be overcome.

Comment 8-10. Figure 2 (and elsewhere): I would not write P(OOH), because that would indicate the presence of a hydroperoxide; rather P(O)OH. // Figure 8 has a bad resolution // Table 2 again contains P(OOH)

Our response: We have updated the figures and tables with better-resolution pictures and corrected chemical formulas in our resubmitted manuscript.

Our Responses to Reviewer #2

Comment 1: The amorphous nature of the synthesized GTUB5-COF posed challenges for X-ray diffraction analysis, which limited the detailed structural understanding of the material. Additional characterization techniques such as X-ray photoelectron spectroscopy (XPS) and advanced electron microscopy are needed. These could provide deeper insights into the chemical states and the morphological details of GTUB5-COF, complementing the findings from NMR and FTIR analyses.

Our response: As suggested, we obtained TEM and XPS data and updated the text accordingly. The combined picture below shows the formation of P-O-P bonds and the degree of crystallinity at higher temperatures. We have shown the formation of P-O-P bonds starting after 130 °C, with the degree of P-O-P bonds increasing with temperature. Until 190 °C, the crystal structure of the COF is similar to the HOF precursor's crystal structure, which we have confirmed using DFT calculations. At 210 °C, some crystallinity is retained, but at 270 °C, the compound becomes completely amorphous. Furthermore, XPS analysis has proven the presence of P-O-P bonds and that the linker cores remain intact at 270 °C.

Comment 2. Given the amorphous nature of GTUB5-COF, computational modeling might offer insights into its probable structure and properties. Molecular dynamics (MD) simulations or density functional theory (DFT) calculations could help understand the gas adsorption capacities and selectivity, as well as the material's stability under various conditions.

Our response: We have performed the DFT studies to get insights into the probable structure and potential packing at low temperatures. As the gas sorption experiments were performed with an amorphous compound obtained at 270 °C, we could not perform the DFT calculations for the comparable gas sorption work. We have now included Prof. Yazaydin from UCL as a contributing co-author.

Comment 3. Perform long-term stability studies of GTUB5-COF under practical operational conditions, including variations in temperature, pressure, and exposure to different gases and liquids.

Our response: We have also performed the gas sorption experiment in the presence of CH_4 and updated the text. We are currently working on detailed stability studies with a different group of PhD and Master students. This ongoing study includes a more comprehensive set of linkers (longer tethered porphyrins, planar polyaromatic linkers, etc.) and is more comparative. Therefore, this work has limited the stability testing to neutral pH, and neutral electrolytes. Furthermore, we have already shown that after repeated water vapor sorption, and CO_2 sorption measurements over 6 months, the P-O-P bonds are still present in the sample.

Comment 4: Conduct a comparative study with other COFs or porous materials targeting similar applications. Such comparisons could highlight the unique advantages or potential drawbacks of GTUB5-COF, providing a clearer context for its practical utility.

Our response: We have now prepared Table 3, which compares the COFs studied in terms of their heat of adsorption and stability.

Comment 5. It is better to elaborate on specific application scenarios or case studies where GTUB5-COF could be particularly advantageous. For instance, detailing its use in a specific industrial gas separation process or in environmental remediation could underscore its practical value and spur interest in further research and development.

Our response: In our resubmission, we have expanded the conclusion section to include potential scenarios of how these compounds can be used in gas separation processes, their potential applications, the possibility of forming thin films, etc.

Comment 6: The synthesis method appears scalable and environmentally friendly, but a detailed cost analysis and scalability study would be beneficial for assessing the potential for industrial applications. Scalability is a critical factor for transitioning from laboratory research to industrial applications, and addressing this could significantly enhance the paper's impact.

Our response: At this stage, we can hypothetically make a kilogram of the linker for ca. 350 €. This is much cheaper than the MOF synthesis using mechanochemistry and continuous flow hydrothermal reactions. The organic synthesis can be further optimized by scaling up processes with buying the starting chemicals at a larger scale from the vendors. TU-Berlin has the patent rights, and we are looking for ways to commercialize these compounds.

Comment 7: Authors should carefully consider their wording, especially in the introduction. For example "The thermal and chemical stability and permanent porosity observed in some COF families opened many potential applications such as gas storage, water adsorption, catalysis, water harvesting, CO₂ capture, catalysis, photocatalysis, semiconductors, energy storage, luminescence."

Our response: We have gone through the article and updated the text. Our changes are highlighted.

Our Responses to Reviewer #3

Comment 1: The main issue with the manuscript is the claim for COF. Although there is no prerequisite of being crystalline to be considered a COF, the common understanding of the field is to expect some degree of crystallinity. Even if a paper reporting new COFs is published without any evidence of order, the community will disregard the findings and also discredit the authors. I think it's important to either reframe the structures as something else or try further to get crystallinity.

Our response: To better explain the P-O-P formation, we have combined the NMR and PXRD data into a single picture (See Below). As it can be seen in the picture below, forming the COF and the degree of P-O-P bonds formed at higher temperatures is a very dynamic process. The P-O-P formation starts at approximately 130 °C, and by 190 °C, the COF phase is already very crystalline with a structure similar to the original HOF molecule. We only observe condensed-diphosphonate groups at this stage, as shown below.

Therefore, it can be clearly stated that up to 190 °C, we retain most of the crystallinity. Above this temperature, the degree of polymerization gradually increases, generating an amorphous compound. At 210 °C, some crystallinity is retained, but at 270 °C, the compound becomes completely amorphous. At 270 °C, the formation of longer P-O-P chains and rings broadens the peaks, generating two broad peaks. We have updated the text and now refer to the compound above 210 °C as a porous polymer rather than a COF. We have also confirmed the presence of the linkers and P-O-P bonds using XPS for the porous polymer phase at 270 °C. Additionally, we have expanded the text and made it clearer to read.

Comment 2. If there is no crystallinity observed, it might not be a bad idea to widen the structure bandwidth. For example, why can't the porphyrin-based monomers condense with each other to make framework structures?

Our response: We are working on strictly planar arylphosphonic acid linkers to improve the crystallinity of the synthesized COFs. We have already formed crystalline thin films and are still working on structural characterization as the thin films have different XRD data compared to the single crystal monomers. We want to submit this work in the upcoming months. We already synthesized new HOFs using only the porphyrin based monomers. These HOFs are being prepared for publication.

Comment 3. The HOF to COF transition experiment shows retention of the grain morphology but not the order. This might be due to the oligomers forming, as the condensation with counter cations must be sterically challenging. The disruption is enough to post disorder but not enough to change the form of the particle.

Our response: Neutralized DMA molecules from the counter HDMA⁺ cations only leave the structure above 200 °C, which may have caused the observed disorder. We have extended the text accordingly and included a discussion on forming P-O-P oligomers. This addition clarifies how the departure of DMA molecules and the steric challenges during the condensation process contribute to the disruption of structural order while retaining the overall morphology.

Comment 4. The CO₂ uptakes are not exceptional. It's probably better to focus on water absorption or H₂S if possible. One evidence is that the H₂O uptake is an order of magnitude higher than CO₂. There are hints on proton conductivity with these materials, but the authors didn't push forward. It might not be a bad idea to focus there.

Our response: The significance of this work is the presence of small pore sizes and increased selectivity for CO₂ in the presence of other gases, such as water vapor and N₂. As suggested, we have now performed CH₄ adsorption studies in addition to CO₂, N₂, and H₂O vapor. We did not observe any CH₄

adsorption. Due to the very small pore sizes of the presented COF, small molecules with a larger kinetic radius than CO₂, such as CH₄ and N₂, cannot enter the pores of the polymer. While H₂O has a smaller kinetic radius than CO₂ and can be adsorbed within the pores, we cannot currently perform H₂S and SO₂ adsorption studies. However, H₂S and SO₂ have larger kinetic radii than CO₂ and N₂, so the polymer is expected to not adsorb them.

We have previously published the proton conductivity of the HOF precursor GTUB5, which exhibited mediocre proton conductivity. As the distance between the neighboring phosphorus atoms is now reduced and the framework has a neutral character, it might provide interesting results. However, we require new humidity and temperature control chambers for proton conductivity measurements.

Our Responses to Formatting Comments

Comment 1. It is hard to navigate with the figures being at the end. If it is revised for another round of reviews, I would prefer seeing them where they first appear.

Our response: We totally agree with the reviewer 3. We also don't like to navigate up and down to see the figures. As we have many authors in this article. It was indeed not manageable to fix the figures into the text after each edit. Therefore, we had to have the figures at the end of the article this time.

Comment 2-5. "using a simple heating in the literature." probably meant "by simple heating." // It is better to put both chemical structures and crystal structures in the Scheme 1. Also, I believe Schemes are not allowed in Nature. // What do you mean by "ca. 2.5 Å (hydrogen bonds with shorter donor-acceptor O-O distances than 2.4 Å)"? 2.5 Å is longer than 2.4 Å? // There are overwhelming details of equipment in the main text. They should be moved to supplementary information.

Our response: We have updated the text based on the formatting suggestions of Reviewer 3 in our resubmitted manuscript.

REVIEWERS' COMMENTS

Reviewer #1 (Remarks to the Author):

The authors have addressed all points I raised. I agree that additional experiments are beyond the scope of the current manuscript. The paper should be published in its revised form.

Reviewer #2 (Remarks to the Author):

The authors have fully addressed the questions, and therefore I recommend it for publication in Nature communications.

Reviewer #3 (Remarks to the Author):

The authors revised their paper following the reviewer comments. The manuscript is much improved and additional characterization data along with description is given for the structural transformation upon heating. However, I'm still having difficulty with the following statement:

“It has a more favorable enthalpy of adsorption value for CO₂ capture at lower pressures than for water vapor, making it a suitable candidate to sequester CO₂ from moist air.”

The structures have a lot higher water uptake than CO₂ at all tested temperatures and concentrations. It's best to delete that bold claim. Without mixed gas (humid gas) uptake study it is purely theoretical.

Response to Reviewer Comments

We would like to give our thanks to all of the reviewers for suggesting the acceptance of our article without further revisions. As reviewer 3 suggested, we removed the sentence “It has a more favorable enthalpy of adsorption value for CO₂ capture at lower pressures than for water vapor, making it a suitable candidate to sequester CO₂ from moist air”

Sincerely,

Gündoğ Yücesan